# Recent Advances in the Recognition Elements of Sensors to Detect Pyrethroids in Food: A Review

**DOI:** 10.3390/bios12060402

**Published:** 2022-06-10

**Authors:** Le Zhang, Mingqi Zhao, Ming Xiao, Moo-Hyeog Im, A. M. Abd El-Aty, Hua Shao, Yongxin She

**Affiliations:** 1Institute of Quality Standards & Testing Technology for Agro-Products, Chinese Academy of Agricultural Sciences, Beijing 100081, China; nkyzhangle@163.com (L.Z.); zhaomingqi1995@163.com (M.Z.); 2Academy of Agriculture and Forestry Sciences, Qinghai University, Xining 810000, China; 1993990035@qhu.edu.cn; 3Department of Food Engineering, Daegu University, Gyeongsan 38453, Korea; imh0119@daegu.ac.kr; 4Department of Pharmacology, Faculty of Veterinary Medicine, Cairo University, Giza 12211, Egypt; abdelaty44@hotmail.com; 5Department of Medical Pharmacology, Medical Faculty, Ataturk University, Erzurum 25240, Turkey

**Keywords:** agriculture, pyrethroids, sensor, recognition elements, recent advances

## Abstract

The presence of pyrethroids in food and the environment due to their excessive use and extensive application in the agriculture industry represents a significant threat to public health. Therefore, the determination of the presence of pyrethroids in foods by simple, rapid, and sensitive methods is warranted. Herein, recognition methods for pyrethroids based on electrochemical and optical biosensors from the last five years are reviewed, including surface-enhanced Raman scattering (SERS), surface plasmon resonance (SPR), chemiluminescence, biochemical, fluorescence, and colorimetric methods. In addition, recognition elements used for pyrethroid detection, including enzymes, antigens/antibodies, aptamers, and molecular-imprinted polymers, are classified and discussed based on the bioreceptor types. The current research status, the advantages and disadvantages of existing methods, and future development trends are discussed. The research progress of rapid pyrethroid detection in our laboratory is also presented.

## 1. Introduction

The withdrawal of highly hazardous organophosphate pesticides from the world market over the last ten years has resulted in pyrethroid insecticides becoming the preferred alternative pesticides due to their effectiveness in pest control [1,2]. Pyrethroids have the broadest application, highest efficiency and lowest residue in addition to their moderate toxicity and biodegradability in plants. To date, more than 70 pyrethroid pesticides have been used in agriculture [3]. Type I pyrethroid pesticides lack α-cyanogen and are represented by permethrin, bifenthrin, and others, while type II structures contain α-cyanogen and are represented by fenpropathrin, cyfluthrin, deltamethrin, and fenvalerate [4,5]. Type II pyrethroid pesticides have good efficacy and stability and are widely used to control pests during crop production.

Pyrethroid pesticides show developmental and neurotoxic effects on mammals and aquatic organisms, which can delay embryo development, increase mortality and the risk of cancer, and even lead to the extinction of aquatic species [6,7,8]. They are also potentially toxic to plants, soil, and aquatic ecology. For example, they are phytotoxic to cucumber seed germination rate, root elongation, branch length, and leaf length [9]. They also disturb soil microbial communities and reduce natural biodegradation [10] and are widely used in household hygiene due to their effective treatment of household pests such as mosquitoes; however, direct contact with pyrethroid pesticides increases health risks [11,12]. Pyrethroid pesticides have strong absorbability and are directly and indirectly transmitted into the food chain, which eventually poses a threat to human health and life [13]. Pyrethroid pesticides are endocrine-disrupting compounds (EDCs) that indirectly interfere with upstream endocrine signal transduction signaling pathways through direct receptor interactions by mimicking and cooperating with endogenous hormones [14]. The pyrethroid metabolite 3-phenoxybenzaldehyde (3-PBA) bioaccumulates in human breast milk, which negatively impacts babies relying on breast milk [15,16]. Although pyrethroid pesticides have no acute toxic effects on humans, long-term exposure may damage male sperm quality and reduce sperm count in F1 offspring during pregnancy and lactation [17,18].

The wide use of pyrethroid insecticides has prompted food safety research to focus on pyrethroid residue detection in crops. Therefore, many countries and organizations have employed strict residue limits for pyrethroid pesticides. For example, the maximum residue level (MRL) of pyrethroid pesticides is in the range of 0.01–4 mg/kg (up to 31 mg/kg for tea) for crops in the European Union; 0.01–20 mg/kg in the USA; 0.01–20 mg/kg (up to 50 mg/kg for hops) in Japan; and 0.01–10 mg/kg (up to 20 mg/kg for tea) in China. Therefore, rapid, sensitive, and effective detection methods should be established to monitor pyrethroid residues in crops and reduce human exposure.

Pyrethroid detection technology includes instrument-based methods [19,20,21,22,23,24] and sensor-based methods. Instrument-based confirmation methods can utilize separation by chromatography [25] combined with strong selectivity and mass spectrometry. This has the advantage of more structural information and high-throughput rapid detection for the accurate analysis of pyrethroid pesticides is possible [25]. The instrument-based method is time-consuming and expensive and requires professional technicians for operation. Therefore, sensor-based methods of agricultural residues have quickly developed. Currently, many recognition elements used to detect pyrethroid pesticides are being used in conjunction with detection techniques. However, few reports have summarized the identification elements and sensor-based methods of pyrethroid pesticides. This review introduces the recognition elements (enzymes, antigens/antibodies, aptamers, and molecular-imprinted polymers) for pyrethroid pesticides and the corresponding determination sensor (SERS or surface-enhanced Raman scattering, SPR or surface plasmon resonance, chemiluminescence sensor, biochemical sensor, fluorescence sensor, and colorimetric sensor) used for detection. Furthermore, the pros and cons associated with sensor-based methods for pyrethroid pesticide detection were analyzed (Table 1).

## 2. Recognition Elements for Pyrethroid Pesticide Detection

Pesticide sensors use biorecognition elements to directly contact the conduction system in space and convert biochemical information into electrical, thermal, optical, and other output forms [38]. These chemical sensors are called biosensors (or biomimic recognition sensors) and can be classified as enzymes, antigens/antibodies, and artificial receptors such as molecularly imprinted polymers (MIPs) and aptamers. They have the advantages of simplicity, rapidity, specificity, high sensitivity, and low cost [39]. The mertis and demertis of the recognition elements for the determination of pyrethroids are summarized in Table 2.

### 2.1. Enzyme-Based Biosensors

Enzyme-based sensors have been studied since 1962 [40] and their sensitivity and universality allow for widespread use in pesticide detection. These are divided into two types of biosensors for pesticide detection: inhibitory and catalytic. Pyrethroids are degraded by the oxidation of P450 monooxygenase, coupling with glutathione S-transferase, and the hydrolysis of phosphotriesterase or carboxyesterase [41]. The hydrolysis of pyrethroids by carboxyl esterase is the primary means of pyrethroid microbial biodegradation. Dongqing et al. described the ester bond catalytic mechanism of carboxylesterase PytH (pyrethroid-degrading carboxylesterase) for pyrethroid pesticides. Carboxylesterases from Sphingobium faniae JZ-2 have α/β hydrolase folding proteins that typically catalyze Ser–His–Asp triples and can effectively hydrolyze many pyrethroid pesticides. PytH has no isomer selectivity compared to other reported pyrethroid-hydrolyzing carboxylesterases, making it a good candidate for pyrethroid residue elimination. Since the hydrolysis efficiency of PytH is relatively low, it can be improved by direct evolution or rational protein design [42]. However, enzymes have strict preservation conditions, poor stability and selectivity, and are inactivated at high organic solvent concentrations [43]. Therefore, to improve their detection performance for pesticides, enzymes are used in conjunction with electrochemical sensors [44].

### 2.2. Antigen/Antibody-Based Biosensors

Immunoassays generally require two steps: hapten construction and antibody preparation. Hapten construction is a prerequisite to obtain the complete antigen of pesticides [45]. Pyrethroid pesticides are hydrophobic molecules and there are therefore great challenges involved in constructing effective haptens for immunoassay methods. Cui et al. [46] synthesized β-cyhalothrin haptens using a one-step method by selectively hydrolyzing the -CN group with a low-toxicity reagent. The haptens were coupled with succinic anhydride-activated carrier protein and the complete antigens were used to prepare polyclonal antibodies (Figure 1). This method replaced the trimethylchlorosilane catalyst with trimethylsilyl trifluoromethane sulfonate to increase the acidity and yield of the preparation to 69%. The method has higher structural fidelity and maintains the integrity of most of the functional groups while modifying the reaction groups. Fruhmann et al. [27] selected and synthesized the immune deltamethrin hapten D133 and cypermethrin hapten C134, established antisera As358-363, and encapsulated the antigens C134-AD, D133-AD, F1-BSA, F2-BSA, F3-BSA, and 3-PBA-BSA. The best combination is As360 (C134-HCH), and the homologous competitor C134-AD can be used for the direct determination of deltamethrin. The IC_50_ was 21.4 ± 0.3 μg/L, and the detection limit was 1.21 ± 0.04 μg/L. The IC_50_ of cypermethrin was 78.6 ± 0.7 μg/L and the limit of detection was 4.56 ± 0.05 μg/L. The antiserum induced by chlorinated compounds was better able to recognize brominated derivatives. This may be because bromine atoms are larger than chlorine atoms and are more suitable for antibody binding sites, which facilitates the antibody recognition of deltamethrin. Alternatively, the difference in electron distribution between chloride and bromide may facilitate recognition. The antibodies prepared by this study are specific to deltamethrin and do not interfere with the determination of other contaminants commonly found in environmental samples by cross-reactivity.

### 2.3. Aptamer-Based Biosensors

Aptamers are single-stranded DNA (ssDNA) or RNA molecules that are synthesized by in vitro chemical methods with high affinity, selectivity, and stability [47]. They consist of 25–30 bases and are systematically optimized for aptamer selection by exponential enrichment using the systematic evolution of ligands by the exponential enrichment (SELEX) technique. Aptamers are more environmentally stable than antibodies and are easier to synthesize in large quantities. Trace targets of ng/kg or even pg/kg can be detected when used in combination with optical and electrochemical techniques [48,49]. However, aptamer development is time-consuming, and complex computational methods can reduce the time and cost by minimizing experimentation [50]. Aptamers are used for pesticide detection by folding ssDNA or RNA into tertiary structures. However, false positive and nonspecific signals may occur due to unpredictable structures and ineffective folding in complex matrices [44]. Yang et al. [48] solved this problem by using a modified capture-SELEX strategy for the selection of λ-cyhalothrin pesticide adaptors. The ssDNA library was fixed to the magnetic bead, and the binding affinity sequence was competitively captured in the magnetic bead after target addition. The recognition mechanism and action site of the aptamer and target were studied using molecular docking technology (Figure 2). A new λ-cyhalothrin colorimetric detection method was established using the aptamer as the recognition molecule and colloidal gold-controlled aggregation mediated by a cationic polymer as the sensing signal. This study provides a reference method for small molecule detection in food. However, aptamers specific to whole-cell SELEX showed significant non-specificity, which may be due to antigen-binding sites on membrane proteins that are ubiquitous in cells [51].

### 2.4. MIP-Based Sensors

MIPs take target molecules [52] or structural analogs [42,43] as templates and functional monomers to synthesize highly cross-linked three-dimensional network structures through covalent or noncovalent crosslinking agents. Template molecules washed with organic solvents leave specific recognition sites in the polymer network that are complementary to template molecules in shape, size, and function [53]. MIPs can be used as a bionic adhesive for biosensors and for grabbing samples during pretreatment. It is superior to antibodies in terms of stability and shelf life, and its target binding performance is equal to or superior to that of natural antibodies [54]. Therefore, MIPs are often used as substitutes for natural antibodies, receptors, and enzymes in biosensors due to their predetermination, recognition and practicability, simple preparation, low cost, and good chemical stability [55] and wide use in environmental monitoring. For instance, Heravizadeh et al. [56] synthesized an MIP with highly specific adsorption for permethrin by precipitation polymerization. The adsorption mass of cis-permethrin and trans-permethrin was up to 7.71 mg, and the method was reliable and effective for the detection of permethrin isomers in biological and environmental samples. Chen et al. [57] used a cyhalothrin template to prepare magnetic MIPs for the rapid high-affinity detection of cyhalothrin in honeysuckle by the precipitation polymerization with an adsorption capacity of 4.9 mg/g. This provides a method to increase MIP binding sites and improve their selectivity. However, using pesticide as the template prevents its complete washing, resulting in template leakage and causing false positive results during detection [58]. Materials with structures and properties similar to those of the target molecule can be used as templates in the synthesis of MIPs to overcome these challenges and improve the application range of MIPs [59]. Cai et al. [60] used phenyl ether as a virtual template to replace pyrethroid pesticides to synthesize molecularly imprinted microspheres (MIMs) and fluorescent tracers (Figure 3). An optimized multiple fluorescence method was then used to simultaneously determine 10 pyrethroid pesticides from 60 real beef and mutton samples with standard recovery rates of 67.77–109%. The advantage of this experiment lies in the virtual template, but its low specificity cannot accurately screen specific pesticides. This problem may be caused by the large MIP surface area, which binds to nonspecific sites such as the sample matrix [61].

## 3. Sensor-Based Methods for Pyrethroid Pesticide Determination

### 3.1. Detection of Pyrethroids Based on Electrochemical Sensors

Electrochemical sensors are a change in current or impedance caused by the combination of a target on an electrode. Changes in the chemical signal modify the electrical signal to quantify pesticides [44]. Esquivel-Blanco et al. [62] developed an electrochemical method for the determination of the pyrethroid metabolite 3-PDB (3-phenoxybenzaldehyde) using laccase as an alternative recognition element. The enzyme is immobilized onto a gold electrode layered with alkanethiol through amide bond formation between the lysine residue of the enzyme and the activated carboxyl group of alkanethiol. The enzyme directly oxidizes the substrate onto the gold electrode with a detection limit of 0.061 μM. In this context, Silva et al. [26] used a polished silver solid alloy electrode (P-AGSAE) related to square wave cathode adsorption stripping voltammetry (SWCASV) as an electrical element for the determination of β-cyhalothrin in water and tea. There was no significant difference compared with gas chromatography–mass spectrometry at the 95% confidence level, proving that the electrochemical method is robust with good stability and sensitivity. Furthermore, Ribeiro et al. [63] used Molegro Virtual Docker-MVD to simulate the docking of four permethrin pesticides at the active site of GST. The results showed that the compounds they selected had high affinity for the catalytic site of the GST enzyme. Therefore, a GST-based screen-printed electrochemical biosensor was developed with LODs of 0.9, 1.6, 3.6, and 9.5 μg/L, respectively, with good accuracy, reproducibility, and stability. Moreover, Borah et al. [64] demonstrated that glutathione S-transferase (GST) immobilized in graphene oxide can be used as an electrocurrent bioelectrochemical sensor for the determination of β-cypermethrin containing 25% methanol, showing that biosensors can be used in relatively high organic solvent concentrations. Additionally, Fruhmann et al. [27] developed an immunosensor based on antigen–antibody measurements and amperometric electrochemical readings that can detect deltamethrin in different water environments by changing electrical parameters, and the limit of detection was 4.7 mug/L in water. MIPs make important contributions to the selectivity and sensitivity of traditional electrochemical methods. If nanomaterials are reintroduced into molecular imprinting, electrochemical sensors will have improved catalytic performance and enhanced conductivity through a rough conductive sensing interface surface [65]. Chansi et al. [66] established a novel immune sensing platform, BSA/Chi-AuNP-rIgG-BSA/MOF/ITO, which used MOF and IgG polyclonal antibody dual screening to detect a variety of pesticides including pyrethroids (Figure 4). The theoretical analysis of rIgG binding was consistent with its functional affinity for a variety of pesticides. Sample detection may be performed with a portable device for simplifying pesticide analysis with minimal heavy metal ion interference, short analysis time, and good stability. Two-dimensional hexagonal boron nitride nanosheets (2D-hBN or white graphene) were used as binding nanomaterials for electrochemical sensors due to their high temperature stability, large surface area, high mechanical strength, and terminal conductivity [67,68]. For instance, Atar and Yola prepared layered nanosheets of an amine-functionalized Fe@AuNPs/2D-hBN nanocomposite electrochemical sensor, which improved the sensitivity of β-cypermethrin detection in wastewater samples based on the synergistic effect of MIPs and nanocomposites [28].

### 3.2. Detection of Pyrethroids Based on Optical Sensors

#### 3.2.1. Surface-Enhanced Raman Scattering Method

SERS is an ultrasensitive vibrational spectroscopy technique used to detect molecules on or near the surface of plasma nanostructures [69]. It has the characteristics of supersensitive, quantitative, real-time detection and multiplexing, and has a wide range of applications in biochemistry and life sciences [70]. The instability of biological components and their insensitivity in identifying small molecule analytes limits their application. When the analyte is adsorbed on the surface of heavy metals, the Raman signal is enhanced, whereas the service life of the SERS substrate is shortened since it does not eliminate the molecules adsorbed on the surface. However, MIP shows high mechanical and chemical stability for small molecule detection, and its molecular selectivity combined with spectroscopy provides a synergistic effect for the fingerprint identification of complex samples [71]. In addition, MIP prevents SERS substrate oxidation and protects the core material [29]. SiO_2_@TiO_2_@Ag@MIPs were designed by Li et al. [29] for the detection of fenvalerate in river water (Figure 5). The use of SiO_2_ prevents the agglomeration of TiO_2_ and Ag and ensures good optical transparency. The composite structure material improves the functionality and selectivity of the SERS substrate, enhances the SERS performance of Ag particles, and degrades the templates adsorbed on its surface. Wang et al. [72] synthesized Fe_3_O_4_/GO/Ag-MIPs (FGA-MIPs) and combined SERS technology to form an FGA-MIP/SERS-imprinted sensor that shows selectivity, good magnetic separation, and the sensitive detection of λ-cyhalothrin in water. This provides a new method for the determination of pyrethroid pesticides in aquatic environments.

#### 3.2.2. Surface Plasmon Resonance Method

SPR is a kind of charge density oscillation with the resonance oscillation of the conduction electron generated by incident light irradiating the interface of a material—such as a dielectric metal film—and the corresponding quantum called a surface plasmon [73]. SPR works by measuring the refractive index near a metal surface, although it cannot distinguish between solutions with the same refractive index. However, modifying the SPR system or modifying the metal film by the active layer or sensing element rectifies this issue (Figure 6) [74].

The combination of small molecule pesticides on the surface of traditional SPR sensors results in a small refractive index with the introduction of nanomaterials enhancing the change in refractive index [75,76]. For example, Liu et al. [30] combined an SPR sensor with Fe_3_O_4_ magnetic nanoparticles (MNPs) by coupling Fe_3_O_4_ and an antibody with MNPs to deliver the carrier of the target analyte to the surface of the sensor. The favorable characteristics of MNPs (large surface area, good magnetism, high refractive index, and high molecular weight) increase the SPR signal, improve sensitivity, and reduce the background interference. This method can simplify sample pretreatment and improve the determination accuracy. The sensitivity of this method for deltamethrin in soybean increases by four orders of magnitude compared with the direct SPR method. In addition, the SPR phase measurement based on the topological properties of the system can replace the amplitude measurement [77].

#### 3.2.3. Chemiluminescence Method

Chemiluminescence sensors can selectively respond to receptor molecules and extract information about specific analytes in complex samples. The optical changes of receptor molecules are of great concern [78]. However, chemiluminescence sensitivity is low and the highly specific recognition of MIPs can play a synergistic role [79]. For example, Zang et al. [80] developed a highly selective chemiluminescence system for fenvalerate by synthesizing fenvalerate MIP using in situ polymerization and applying the quenching mechanism of the luminol–H_2_O_2_–NaOH chemiluminescence system. A chemiluminescence method for fenpropathrin detection was developed in a similar way by Zhao et al. [81] in the same laboratory. This method improved the adsorption performance of MIP and enhanced the enrichment performance of fenpropathrin compared with the chemiluminescence method by synthesizing double-sided hollow MIP microspheres. However, the special Y-shaped tubes used in these two articles meant that only fenvalerate and fenprothrin were determined instead of multiple simultaneous detections. Huang et al. [31] prepared an MIP that identifies 10 pyrethroid pesticides using double virtual templates and designed a chemiluminescence sensor for the determination of chicken pyrethroid pesticides. This method was repeatable four times, had a detection time of 12 min, and the limits of detection were in the range of 0.3–6.0 pg/mL in the 10 analytes.

#### 3.2.4. Fluorescence Method

A fluorescent biosensor measures analytes by fluorescence enhancement or quenching caused by direct interaction between the fluorescent probe and analyte [82]. Current fluorescent substances used for analysis and determination include quantum dots, carbon dots, rare earth elements, and fluorescent dyes [83].

The fluorescence detection method has high efficiency, simplicity, and sensitivity and may be combined with an MIP for specific recognition and enrichment [33]. Samples are detected by fluorescence quenching after the target substance binds with MIP. Wang et al. [33] showed that the 5(6)-isothiocyanate (FITC) and 3- aminopropyltriethoxysilane (APTS)/SiO_2_ composite fluorescent MIP selectively recognizes and detects λ-cyhalothrin. This method eliminates the interfering substances in the sample and improves the detection limit.

*Quantum dots*. QDs are a new type of semiconductor fluorescent nanocrystal with a high quantum yield and narrow emission spectrum. Narrow photoluminescence bands caused by quantum dots provide bright light even for individual molecules [65]. Most quantum dots are synthesized by toxic, unstable heavy metals such as cadmium, which potentially harm the environment and organisms [84]. Therefore, it is necessary to passivate the shell to reduce heavy metal leakage. Quantum dots can be protected by coupling them with enzyme-, antibody-, and MIP-based nanomaterials on their surface [85]. For example, MIP-QDs obtained after the surface functionalization of quantum dots show high selectivity and fluorescence characteristics for the target and may be used for pesticide detection [86,87,88]. Li et al. prepared a novel eco-friendly MIP-QD sensitive fluorescence nanosensor for the selective quenching fluorescence of cyfluthrin based on FeSe-QDs using an optimized reverse microemulsion method (Figure 7). The specific recognition of cyfluthrin is due to ionic interactions, molecular structure selection and hydrogen bond interactions to prevent charge transfer from FeSe-QDs to cyfluthrin, resulting in the phenomenon of fluorescence quenching. This method has excellent linearity, selectivity, and sensitivity, and was used for detecting cyfluthrin in fish samples [34].

*Carbon dots*. Fluorescent carbon dots (CDs), also known as carbon quantum dots, are novel, zero dimensional (0D; diameter below 10 nm), nontoxic photoluminescent carbon nanomaterials [89]. They are used as a fluorescence response signal due to their strong luminescence and controllable performance [90,91]. They are widely studied because of their chemical stability, good electrical conductivity, biocompatibility, luminescence, and wide absorption wavelength range [92,93,94]. In recent years, many studies have applied CDs to provide technical guidance for pollutant detection in the environment [95]. In most cases, the fluorescence quenching of solid CDs occurs [96] but their combination with titanium, nickel, and cadmium enhances light absorption and the visible light response, which improves their photocatalytic performance [97]. Although fluorescent CDs can be directly used to detect analytes, their sensitivity and anti-interference ability are low. However, MIPs can compensate for this defect, and combination with CDs to prepare fluorescent nanomaterials for pesticide detection is possible [98,99,100].

Zhu et al. [35] developed a simple and effective two-channel specific fluorescence method for the determination of λ-cyhalothrin based on dual-emission blue–green CD functionalized core–shell nanospheres. The blue and green CDs were taken as the reference signal and the detection signal, respectively. The use of ionic liquids with a wide viscosity range and good stability can improve the detection sensitivity and selection range of core–shell nanospheres. Combination with smartphone integrated optics enables the real-time detection of λ-cyhalothrin by monitoring the fluorescence changes from green to blue. The interference of shortwave backgrounds was overcome by the introduction of functional groups resulting in the development of a red-emitting carbon point (RCD) with stable emission characteristics wherein the CD moves in the direction of red wavelengths [101]. Zhu et al. prepared an on-site visualization and rapid detection RCD MIP sensor for the quantitative detection of λ-cyhalothrin using a smartphone (Figure 8) [102].

*Time-resolved fluorescence microsphere*. Ordinary fluorescent groups are easy to quench during detection, and the detection time is greatly reduced (Stokes shift is 1–100 nm) due to high photobleaching and chemical degradation efficiency [103]. Trivalent rare earth ions such as Eu(III), Tb(III), and Sm(III) were used as labels in time-resolved fluorescence analysis and may be used as a fluorescence quantitative immunoassay. The Stokes shift and fluorescence lifetime were above 150 nm and up to 1 millisecond, respectively [104]. Lanthanide chelates produce high-intensity fluorescence, strongly resist photobleaching, and have a long decay time (1250 μs). This delays the measurement time and eliminates the interference of natural fluorescence, which greatly improves method sensitivity [105,106,107]. Rare earth-based nanomaterials are stable and bright fluorescent probes, so they are often the best nanoprobes for hypersensitive biological detection [108]. They are typically used to detect agricultural residues through the preparation of an immunochromatographic strip [109,110,111,112]. However, we did not find relevant literature in the Web of Science containing time-resolved fluorescence and pyrethroid pesticides as keywords.

### 3.3. Detection of Pyrethroids Based on Biosensors

#### 3.3.1. Biochemical Method

The use of antibodies and enzymes as recognition elements has the advantage of high throughput, convenience, good sensitivity, and simplicity [113]. Typically, enzyme-linked immunoassays (ELISAs) are employed. The enzyme has very high sensitivity and can be used for the trace detection of pesticides with a detection limit of 10^−10^ M. However, the short lifetime of the enzyme and the interference of impurities such as metal ions in the matrix result in weaker specificity and limit its application [46,114]. Although ELISA has good detection sensitivity, there may be technical difficulties during pesticide labeling [115]. López Dávila et al. [116] used the Abraxis pyrethroid assay kit to determine permethrin in water (control group), cucumber, tomato, and bell pepper and determined a minimum detectable concentration of 10 μg/L using the Log10 value of B/B_0_% as the *Y* axis and the permethrin concentration as the *X* axis. The cross-reaction test of 12 pesticides showed consistent results with gas chromatography–electron capture detection (GC-EDC) and ultra-performance liquid chromatography–tandem mass spectrometry (UPLC–MS/MS). This study became the basis for the Cuban pesticide residue detection program ELISA kit. Huo et al. [117] developed a fast and sensitive direct competitive fluorescein immunoassay (DC-FEIA) to detect the pyrethroid metabolite 3-PBA based on a nanobody (Nb)-alkaline phosphatase (AP) fusion protein. The IC_50_ of this method is nearly ten times higher than that of direct ELISA and can detect 3-PBA in urine. Xiao et al. [32] established an ELISA method to detect 0.05–620 mg/kg cypermethrin fish based on MIP-QDs (MIP-quantum dots) (Figure 9). The method shows linear fluorescence quenching and combines the advantages of rapid, sensitive, and efficient ELISA with the high specificity and sensitivity of MIP-QDs.

#### 3.3.2. Colorimetric Method

Some colorimetric signals can be observed with the naked eye or read with a smartphone. Although colorimetric methods are easy to prepare and enable rapid detection, most food extracts are colored, which interferes with detection [118]. Colorimetric reaction methods are mostly based on membranes and paper or microfluidic chips [119]. Immunochromatography is a colorimetric analysis method that combines immunoassays with chromatography. It is widely used for monitoring agricultural products because of its fast detection, strong specificity, and lack of requirement for professional instruments or specialized staff for operation, in contrast to ELISA [120]. Therefore, market regulators and ordinary consumers can instantly detect pesticide residues in agricultural products. In addition, immunochromatography can detect pyrethroids within 10 min, which is much faster than ELISA.

Most experiments involving small molecule antigens with single epitopes—such as pesticides and veterinary drugs—are designed and explored by the competition method. Meanwhile, large molecule antigens with multiple epitopes, such as proteins and toxins, are determined by the sandwich method. A traditional immunochromatographic technique involves labeling colloidal gold [121] or fluorescent substances [122] on the monoclonal antibody to conjugated antigen at the test line. By reading the values of the test line and control line, the results can be qualitatively and quantitatively judged. However, antibody labeling with colloidal gold results in low sensitivity compared with labeling using fluorescent substances [123]. Costa et al. [124] developed a silicon dioxide-coated mesoporous material to selectively identify type I pyrethroids based on lateral-flow strips. The analyte can be detected in 2 min with a limit of detection of 1 ppb using signal readings from smartphones. Although this method can quickly detect permethrin with high sensitivity, the experimental design is complicated and unsuitable for large-scale use. Li et al. [36] established an immunochromatographic method for the determination of cypermethrin and fenvalerate using two test lines(Figure 10). Competitive interference between the different pesticides was prevented by coating the two test lines with two types of haptens with qualitative analysis observed from the two color changes. The method determined dual pesticides in tap water, river water, and milk with the data analyzed by the 2plex-speclysis application. SERS technology was used for the quantitative analysis of the tested pesticides with the following limits of detection for cypermethrin and fenvalerate: 2.3 × 10^−4^ and 2.6 × 10^−5^ ng/mL, respectively. In addition, the data analysis method is customized and can be used by nonprofessionals.

A typical successful case involving the combination of a colorimetric biosensor and a fluorescent biosensor is fluorescence immunochromatography. This enables sensors on antibodies to fluoresce under certain excitation wavelengths. Fluorescein is commonly used in immunoassay methods, including fluorochrome [125], carbon dots [126], and lanthanide series [127]. For example, Zhao et al. [37] developed smartphone-based dual-channel immunochromatographic strips (ICTS) to synthesize carbon point PCD with ultrahigh fluorescence brightness as a signal amplifier to simultaneously detect cypermethrin and its 3-PBA metabolite. Images were analyzed and recorded on smartphone devices according to the red fluorescence obtained. This has potential in the direct detection of pyrethroid insecticides. However, it is difficult to produce antibodies against pesticides in animal bodies due to their small molecular structure, long antibody preparation period, harsh preservation conditions, and the need for sacrificing animals. Therefore, it is necessary to develop a highly specific bionic recognition material to replace animal antibodies and combine immunological methods for the rapid determination of pyrethroid pesticides.

MIPs can be used as a biomimetic material for the rapid and specific identification of pyrethroid pesticides when combined with immunofluorescence technology. However, if the lateral flow chromatography strip is designed according to the preliminary study of our laboratory, the free MIP is directly fixed on the nitrocellulose membrane of the strip, and the polymer will elute with the chromatography liquid, resulting in inaccurate results. Therefore, a substance that can be used as a carrier to fix the polymer on the nitrocellulose membrane as a material for antigen recognition is required for immunochromatographic analysis. The combination of molecularly imprinted electrospinning identification material can replace the use of monoclonal antibodies in immunoassays. For example, He et al. [128] established a method to determine triazolin in water based on a molecularly imprinted biomimetic immunofluorescence strip from our laboratory (Figure 11). In this method, triazolin MIP was fixed on the strip by an electrospinning membrane to replace antibodies. All water samples were negative, which was consistent with the LC–MS/MS results. This experiment provides a worthy idea for studying immunological methods by combining molecularly imprinted electrostatic bionic materials with fluorescence immunochromatography.

Our laboratory improved the sensitivity of the method while reducing background interference by using time-resolved fluorescent latex microspheres labeled with fenvalerate hapten-IgG as a fluorescent probe in immunochromatography for the detection of pyrethroid pesticide residues. This study is based on a competitive experimental design. When fenvalerate is present in the sample, it competes with the fluorescent probe for the specific binding site of MIP and causes fluorescence signal changes. These ideas were used to construct a new bionic fluorescent immunochromatographic strip for the detection of pyrethroid pesticides.

## 4. Conclusions and Future Perspectives

The use of pyrethroids in agriculture and health is increasing; therefore, rapid detection has become the mainstream research direction. Sensors that specifically identify pyrethroids are becoming more popular. Most pyrethroids are optical isomers containing chiral carbon atoms with similar structures. Therefore, the immunoanalysis cross-reaction rate is high, and the pesticide is a small molecule that results in the time-consuming preparation of traditional monoclonal antibodies. In comparison, bionic identification materials can solve the technical problems encountered during monoclonal antibody preparation and may replace animal antibodies in the future. However, low sensitivity is a key problem that urgently needs an breakthrough. The use of fluorescent material for labeling can improve the detection sensitivity. The combination of a variety of technologies may further improve the rapid detection of pyrethroid pesticides, which is of great significance for agricultural planting and public health.

## Figures and Tables

**Figure 1 biosensors-12-00402-f001:**
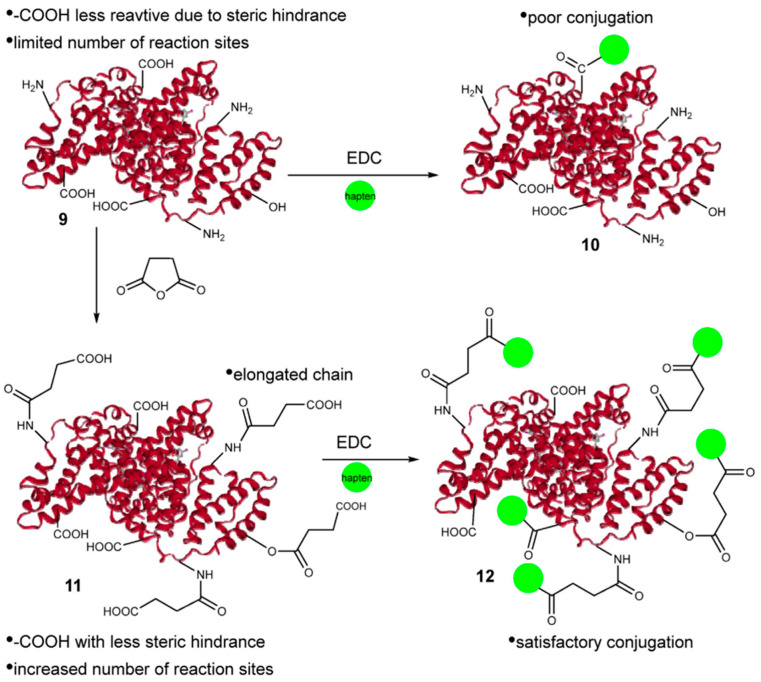
Illustrative figure of the succinic acid treatment and conjugation process ([46]).

**Figure 2 biosensors-12-00402-f002:**
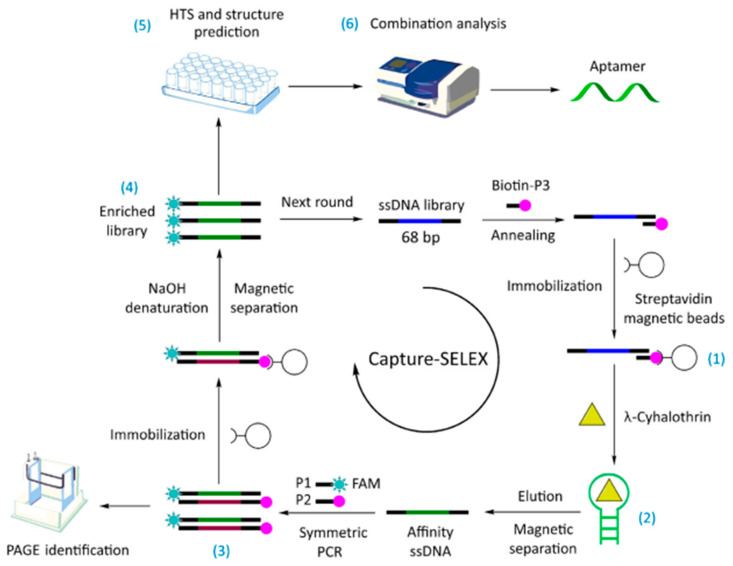
The capture-SELEX technology roadmap for screening aptamers bound to l-cyhalothrin ([48]). (1 Immobilization of random ssDNA library; 2 Elution of affinity ssDNA sequences; 3 Symmetric PCR amplification and electrophoresis identification; 4 Preparation of ssDNA enrichment library; 5 High-Throughput Sequencing (HTS) and secondary structure prediction; 6 Binding characteristics of affinity sequences).

**Figure 3 biosensors-12-00402-f003:**
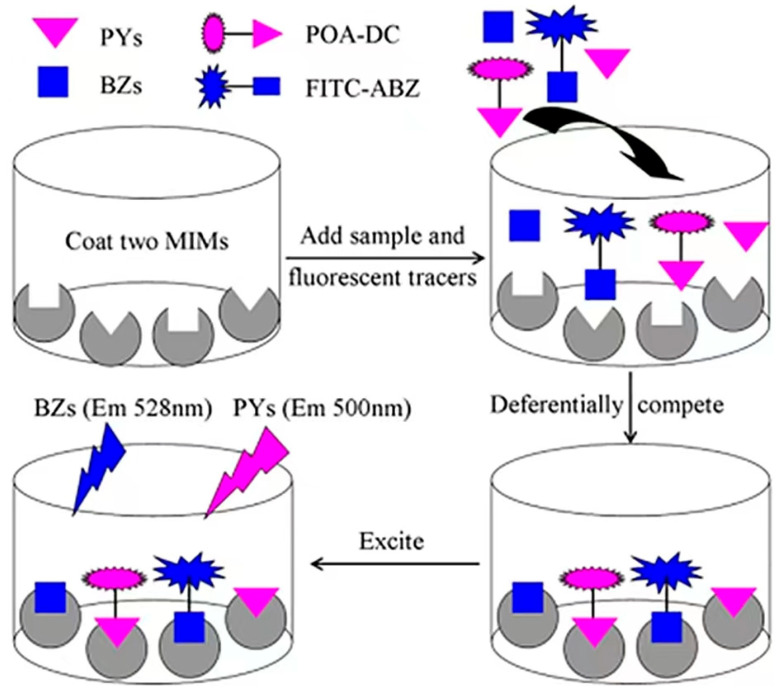
Assay principles of the MIM-based multiplexed fluorescence method ([60]).

**Figure 4 biosensors-12-00402-f004:**
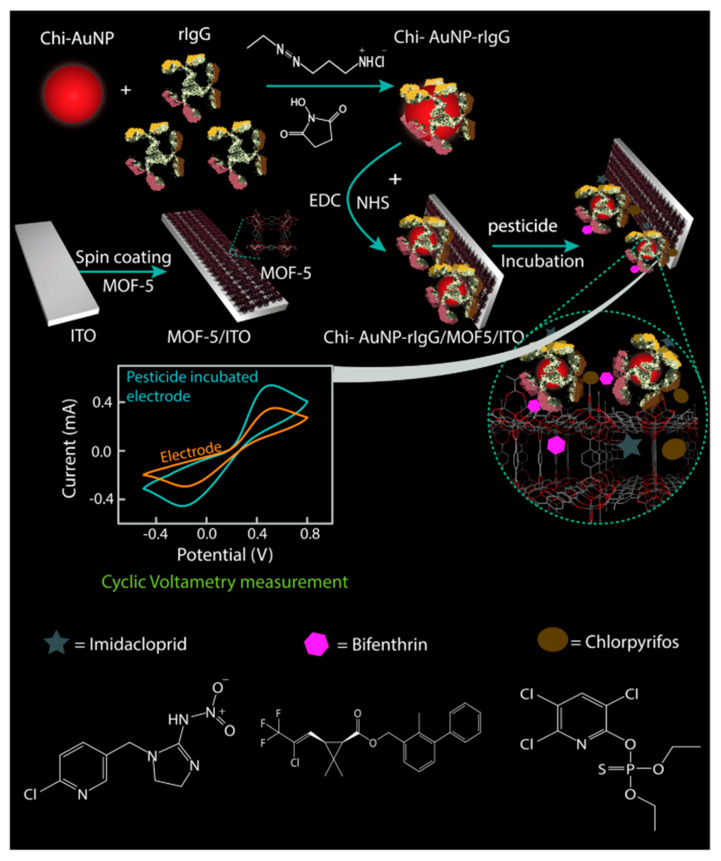
Schematic representation of the mechanism of fabrication and detection of pesticides by immunoelectrode ([66]).

**Figure 5 biosensors-12-00402-f005:**
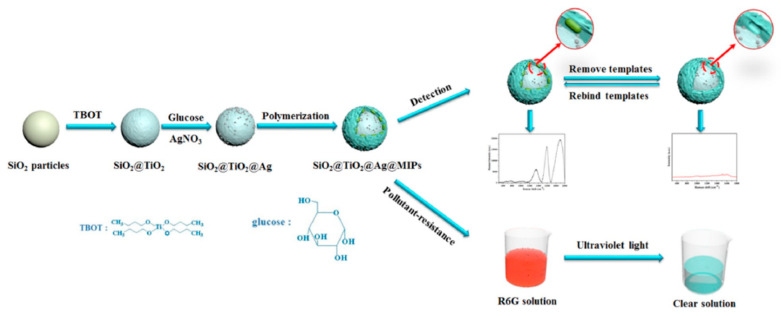
Illustration of the preparation of SiO_2_@TiO_2_@Ag@MIPs and SERS detection of FE upon specific recognition ([29]).

**Figure 6 biosensors-12-00402-f006:**
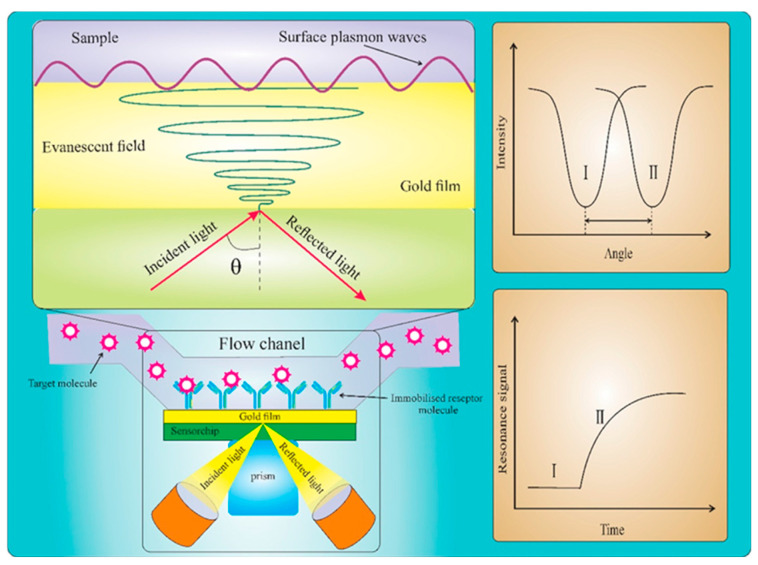
A schematic illustration of the conventional Kretschmann optical configuration for SPR biosensing and the associated angle shift and sensorgram plot of the resonance signal change with time ([67]).

**Figure 7 biosensors-12-00402-f007:**
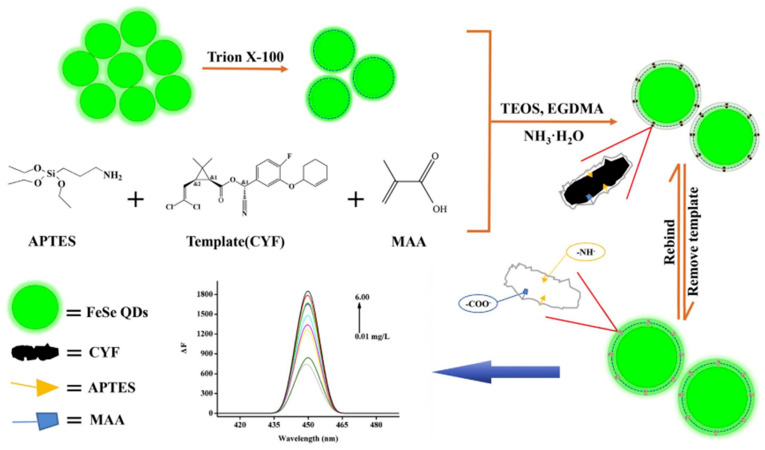
Schematic representation of a fluorescent nanosensor based on MIP-FeSe-QD ([34]).

**Figure 8 biosensors-12-00402-f008:**
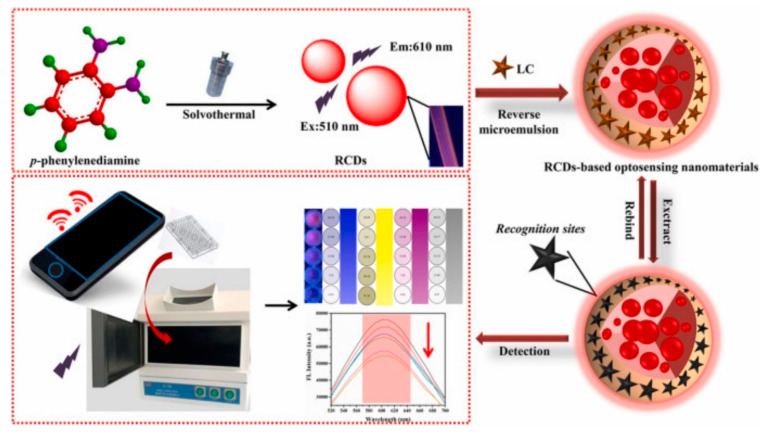
Schematic illustration of the platform comprising RCD-based optosensing nanomaterials for the detection of LC ([102]).

**Figure 9 biosensors-12-00402-f009:**
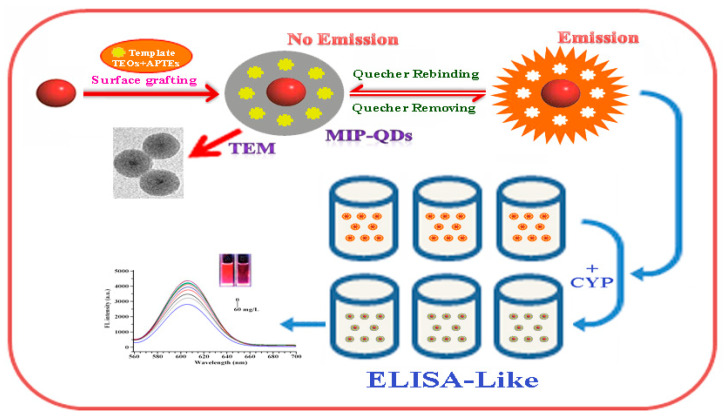
An ELISA-like method based on the MIP-QDs to monomer cypermethrin in the samples ([32]).

**Figure 10 biosensors-12-00402-f010:**
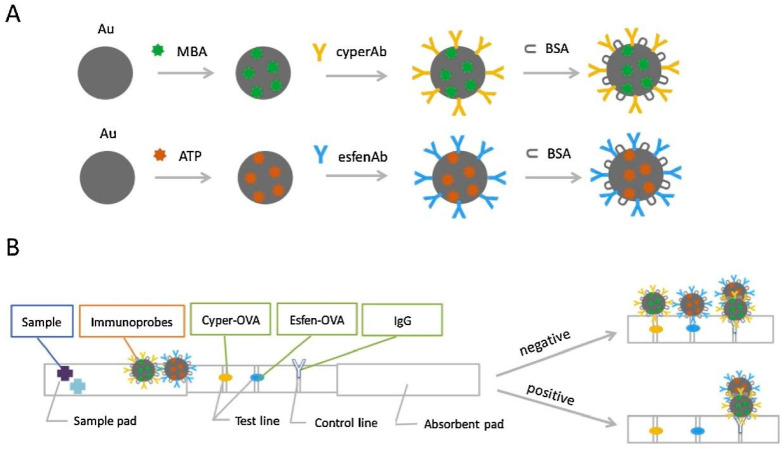
(**A**) Schematic illustration showing the preparation of two types of immunoprobes Au-MBA-cyperAb and Au-ATP-esfenAb; and (**B**) assembly of the ICA-SERS strip and schematic diagram of the mechanism for multiplex detection ([36]).

**Figure 11 biosensors-12-00402-f011:**
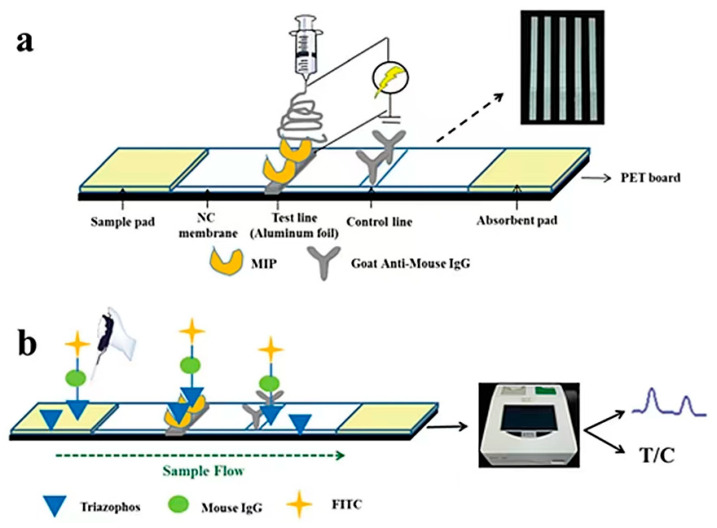
(**a**) Structure diagram of the molecularly imprinted electrospun test strip and (**b**) direct competitive fluorescence detection process ([128]).

**Table 1 biosensors-12-00402-t001:** Representative examples of pyrethroid residue detection based on sensors.

Pyrethroid	Recognition Element	Reading Device	Linear Range	LOD	Sample	Mertis	Reference
β-cyhalothrin	P-AGSAE	SWCASV	3.0 × 10^−6^–1.0 × 10^−5^ mol/L	8.1 µg/L	water, tea	high robustness, good stability, sensitivity	Silva et al. [26]
Deltamethrin	ELISA	amperometric biosensor	-	4.7μg/L	seawater	without any pretreatment	Fruhmann et al. [27]
cypermethrin	amine-functionalized Fe@AuNPs/2D-hBN	molecular imprinted sensor	1.0 × 10^−13^–1.0 × 10^−8^ M	3.0 × 10^−14^ M	wastewater	stability, repeatability	Atar and Yola [28]
fenvalerate	SiO_2_@TiO_2_@Ag@MIPs	SERS	1.0–100 nmol/L	0.2 nmol/L	river water	functionality, selectivity, self-cleaning	Li et al. [29]
deltamethrin	Fe_3_O_4_-MNPs	SPR	0.01–1 ng/mL	0.01 ng/mL	soybean	increases the SPR signal, improves sensitivity, low background interference	Liu et al. [30]
10 pyrethroids	MIP	chemiluminescence sensor	0.3–6.0 pg/mL	-	chicken samples	short detection time, repeatable	Huang et al. [31]
cypermethrin	ELISA-like method	MIP-QDs	0.05–60.0 mg/kg	1.2 μg/kg	fish	rapid, sensitive, high specificity, sensitivity	Xiao et al. [32]
λ-cyhalothrin	SiO2@FITC-APTS@MIPs	fluorescence quenching	0–60 nm/L	9.17 nM/L	Chinese spirits	good monodispersity, high fluorescence intensity, good selective recognition	Wang et al. [33]
cyfluthrin	FeSe-MIP-QD	fluorescence quenching	0.010–0.20 mg/L	1.0, 1.3 µg/kg, respectively	fish, sediment samples	selectivity, sensitivity	Li et al. [34]
λ-cyhalothrin	blue and green CDs	ratiometric fluorescence core–shell nanosphere	1–150 mug/L	0.048 mug/L	tap water, tea, cucumber, apple	sensitivity and selection range	Zhu et al. [35]
cypermethrin, fenvalerate	immunochromatographic assay (ICA)	SERS	10^−5^–100 ng/mL	2.3 × 10^−4^, 2.6 × 10^−5^ ng/mL	tap water, river water, milk	simple, sensitive, nonexpert people	Li et al. [36]
cypermethrin, 3-PBA metabolite	dual-channel immunochromatographic test strip (ICTS)	smartphone	1–100 ng/mL, 0.1–100 ng/mL, respectively	0.35 ng/mL/0.04 ng/mL	standard sample	low cost, high sensitivity, and simple operation	Zhao et al. [37]

**Table 2 biosensors-12-00402-t002:** The advantages and disadvantages of the recognition elements for the determination of pyrethroids.

Sensors	Example	Mertis	Demertis
enzymes	P450 monooxygenase, GST, phosphotriesterase, carboxyesterase	sensitivity, universality	strict preservation conditions, poor stability and selectivity, inactivated at high-organic solvent concentration
antigens/antibodies	-	specificity, high specificity	sacrifice animals, long experimental period, difficult-to-construct haptens
aptamers	single-stranded DNA (ssDNA) or RNA	high affinity, selectivity, stability, environmentally stable, easier to synthesize	time-consuming, unpredictable structures, ineffective folding, non-specificity
chemical synthesis	MIP	predetermination, recognition, practicability, simple preparation, low cost, good chemical stability	non-specificity

## Data Availability

This article has not been submitted to other journals, and the cited materials are labeled references.

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
