# Peer review of "Recent Advances in the Recognition Elements of Sensors to Detect Pyrethroids in Food: A Review"

_biosensors, 2022, doi:10.3390/bios12060402_

Round 1

Reviewer 1 Report

This mini-review is interesting but needs major revision. Please find my comments below:

1. Please revise English throughout the article. For example:

“Therefore, enzymes are combined with other technologies to improve their performance for pesticide detection , for example, bind to electrochemical sensors[32]. At present, there are few enzymes used for the determination of pyrethroid pesticides, so this 102 review focuses on the following aspects.”

2. Please improve the resolution of all figures.

3. Please revise the structure/sections of the review article since it is confusing. For example the authors start with section 1. Introduction and end with 1.14 Conclusions and future perspectives???

4. The authors mention in the Introduction section that they will discuss about the problems associated with rapid detection technology for pyrethroid pesticide. Unfortunately, it is not very clear for me where they do this throughout the article.

5. The authors omitted some of the new articles from literature that might be relevant for this mini-review manuscript. Also, sometimes tables are important to synthesize or compare information.

For example:

- Regarding the recognition elements for pyrethroid detection the authors could include aslo the whole-cell biosensors (Sci Rep 9, 12466, 2019 doi:10.1038/s41598-019-48907-6)

- Regarding the enzyme-based sensors they could include more recent publications (Sensors and Actuators Reports 4, 2022, doi: 10.1016/j.snr.2022.100093)

- There are many immunoassays developed for pyrethroids detection. A table containing few examples might be relevant/interesting for the reader.

- What about the biomimetic ELISA (BELISA) or biomimetic nanozyme-linked immunosorbent assay (BNLISA) for pyrethroid detection?

Author Response

Thank your, we have read your comments and suggestions on our manuscript. Please see the attachment.

Reviewer 2 Report

-A table showing different (bio)sensors would be useful in comparing the various detection methods. This should include the type of pyrethroid, recognition element used, type of detection and the performance parameters (linear range, limit of detection….) and other elements if necessary.

-The structure of the review should be changed to be clearer. In the current form it has 1. As Introduction,  1.1. as Recognition elements, 1.2.-1.5. lists then 1.6. is for "Rapid methods", the  1.7-1.13 discuss detection methods based on techniques, and the Conclusions as 1.14.

Perhaps it could be changed to:

1.Introduction

2. Recognition elements

2.1. Enzymes

….

3. Classification based on techniques

3.1. Electrochemical detection methods

3.2. SERS-based method

4. Conclusions

Or a similar structure.

"1.7. Rapid detection method based on an electrochemical biosensor" also seems to be incorrect. Bellow there are the optic methods such as SERS and SPR. The previous section 1.6. had the electrochemical methods.

-The text should also explain in more detail the phrase "Rapid methods". What are the assay times for the methods mentioned in the review.

-The text contains fragments such as "Enzyme-based sensors", that should be rephrased as "Enzyme-based biosensors". These should take into account the classification of sensors and biosensors.

-The Conclusion section should mention more about the different methods that were listed and their performance, about which type of recognition elements and surface modification techniques (of nanomaterials for example) showed more promise.

-The work talks about "Biological sensor recognition elements". In the literature there are already many articles talking about MIP-based sensors as "biosensors". However, in my opinion this is not correct as MIPs are entirely synthetic recognition elements, not biological ones. Aptamers, for example, are synthetic but still based on nucleotides which are biomolecules, MIPs are not. Therefore, it would be more correct to rephrase the work as talking about "recognition elements" in general. This is only a recommendation, to be implemented if the authors agree and see fit. 

Author Response

(The authors gave the same response as above.)

Reviewer 3 Report

Ref: biosensors-1738182

Title of the manuscript: “Biological sensor recognition elements to detect pyrethroids in food: A review.”

In this paper, Zhang et al. complied the biosensor recognition elements for the detection of pyrethroids in the food systems. The review is indeed interesting and aptly written. Therefore, I recommend the acceptance of this manuscript after properly addressing the below points.

  1. There are a few places which are with the alphabet ‘X’. It might be some error or reference needed to be added at suitable places. There is no reference cited for line 336. Please check Line 90, 240, and 412.
  2. There are a few grammatical errors. Please check lines 188 and 350.
  3. Figures must be of high quality. The text in the figures should be visible and clear. Check Figures 5, 6, 7, and 11.
  4. Introduce the required abbreviation in the manuscript when required. Example ‘MIP’ (Line 162), ms (Line 379), MIM should be introduced as ‘molecularly imprinted microspheres’ before adding it in the figure caption (Line 91).
  5. Please check the units for detection limit ‘mm’ (Line 195).

Author Response

(The authors gave the same response as above.)

Round 2

Reviewer 1 Report

Dear authors,

My suggestion to include/mention/discuss in the manuscript about the whole-cell biosensors or about the biomimetic nanozyme-linked immunosorbent assay for pyrethroids detection was in order to improve your manuscript and in the future to attract more citations, since both are hot topics in the field nowadays. Unfortunately, you were not willing to do that and to be honest I am not sufficient satisfied by your responses.  Anyway, I will not insist on this aspect.